

# The ESA MIPAS/ENVISAT Level2-v8 dataset: 10 years of measurements retrieved with ORM v8.22

Bianca Maria Dinelli[1,*], Piera Raspollini[2,*], Marco Gai[2,*], Luca Sgheri[3], Marco Ridolfi[4], Simone Ceccherini[2], Flavio Barbara[2], Nicola Zoppetti[2], Elisa Castelli[1], Enzo Papandrea[1], Paolo Pettinari[1,5], Angelika Dehn[6], Anu Dudhia[7], Michael Kiefer[8], Alessandro Piro[9], Jean-Marie Flaud[10], Manuel López-Puertas[11], David Moore[12], John Remedios[12], and Massimo Bianchini[13]

[1]CNR-ISAC – Istituto di Scienze dell'Atmosfera e del Clima del Consiglio Nazionale delle Ricerche, Via Gobetti, 101 – 40129 Bologna, Italy
[2]CNR-IFAC – Istituto di Fisica Applicata "Nello Carrara" del Consiglio Nazionale delle Ricerche, Via Madonna del Piano, 10Via Madonna del Piano,10 – 50019 Sesto Fiorentino (FI), Italy
[3]CNR-IAC – Istituto per le Applicazioni del Calcolo Mauro Picone del Consiglio Nazionale delle Ricerche, Via Madonna del Piano, 10 – 50019 Sesto Fiorentino (FI), Italy
[4]CNR-INO – Via Madonna del Piano, 10 – 50019 Sesto Fiorentino (FI), Italy
[5]Dipartimento di Fisica e Astronomia - Universita' di Bologna, Bologna, Italy
[6]ESA-ESRIN, Frascati, (Rome), Italy
[7]Atmospheric, Oceanic and Planetary Physics, University of Oxford, Clarendon Laboratory, Parks Road, Oxford OX1 3PU, UK
[8]Karlsruhe Institute of Technology, Institute of Meteorology and Climate Research, Karlsruhe, Germany
[9]Serco Italia S.p.A., Frascati, (Rome), Italy
[10]LISA- Laboratoire Interuniversitaire des Systèmes Atmosphériques (LISA), UMR CNRS 7583, Université de Paris et Université Paris Est Créteil, Institut Pierre Simon Laplace (IPSL), 61 Avenue du Général de Gaulle, F-94010 Créteil Cedex, France
[11]Instituto de Astrofísica de Andalucía (IAA-CSIC), Glorieta de la Astronomía s/n, 18008 Granada, Spain
[12]University of Leicester, Physics and Astronomy, Leicester, UK
[13]CNR-ISC – Istituto dei Sistemi Complessi del Consiglio Nazionale delle Ricerche, Section of Florence, Italy
*These authors contributed equally to this work.

**Correspondence:** B. M. Dinelli (bm.dinelli@isac.cnr.it)

**Abstract.** The observations acquired during the full mission of the Michelson Interferometer for Passive Atmospheric Sounding (MIPAS) instrument, on board the European Space Agency ENVISAT satellite, have been analysed with version 8.22 of the Optimised Retrieval Model (ORM), originally developed as the scientific prototype of the ESA level 2 processor for MIPAS observations. The results of the analyses have been included into the MIPAS level 2 version 8 (level2-v8) database containing atmospheric fields of pressure, temperature and volume mixing ratio of MIPAS main targets $H_2O$, $O_3$, $HNO_3$, $CH_4$, $N_2O$, and $NO_2$, along with the minor gases $CFC-11$, $ClONO_2$, $N_2O_5$, $CFC-12$, $COF_2$, $CCl_4$, $CF_4$, $HCFC-22$, $C_2H_2$, $CH_3Cl$, $COCl_2$, $C_2H_6$, $OCS$, $HDO$. The database covers all the measurements acquired by MIPAS in the nominal measurement mode of the Full Resolution (FR) part of the mission (from July 2002 to March 2004) and all the observation modes of the Optimised Resolution (OR) part (from January 2005 to April 2012). The number of species included in the MIPAS level2-v8 data-set



makes it of particular importance for the studies of stratospheric chemistry. The database is considered by ESA the final release of the MIPAS level 2 products.

The ORM algorithm is operated at the vertical grid coincident to the tangent altitudes of the observations or to a subset of them, spanning (in the nominal mode) the altitude range from 6 to 68 km in the FR phase and from 6 to 70 km in the OR period. In the latitude domain, FR profiles are spaced by about 4.7 degrees while the OR profiles are spaced by about 3.7 degrees. For each retrieved species the auxiliary data and the retrieval choices are described. Each product is characterised in terms of the retrieval error, spatial resolution, and 'useful' vertical range in both phases of the MIPAS mission. These depend on the characteristics of the measurements (spectral and vertical resolution of the measurements), on the retrieval choices (number of spectral points included in the analyses, number of altitudes included in the vertical retrieval grid), and on the information content of the measurements for each trace species. For temperature, water vapour, ozone and nitric acid the number of degrees of freedom is significantly larger in the OR phase than in the FR one, mainly due to the finer vertical measurement grid. In the FR phase some trace species are characterised by a smaller retrieval error with respect to the OR phase, mainly due to the larger number of spectral points used in the analyses, along with the reduced vertical resolution. The way of handling possible caveats (negative VMR, vertical grid representation) is discussed. The quality of the retrieved profiles is assessed through four criteria, two providing information on the successful convergence of the retrieval iterations, one on the capability of the retrieval to reproduce the measurements, and one on the presence of outliers. An easy way to identify and filter the problematic profiles with the information contained in the output files is provided. MIPAS level2-v8 data are available to the scientific community through the ESA portal https://earth.esa.int/eogateway/.

# 1 Introduction

Satellites for the Earth observation have produced and are producing an enormous volume of measurements, improving our knowlegde of the Earth's system and climate. Although our picture of the Earth's atmosphere is getting more and more accurate over the years, there still is the need to study long term variations of its state, especially in the stratosphere and in the Upper Troposphere-Lower Stratosphere (UTLS), and their influence on climate.

The global and multi-year coverage of satellite missions is producing a detailed picture of the atmosphere, through the measured distribution, variability and long term trends of its physical and chemical state. Among the numerous observation techniques, infrared limb observations, that can be performed in absorption, exploiting the Sun or the Stars as infrared source (solar or stellar occultation technique), or in emission, exploiting the thermal emissions of the atmospheric molecules, have the peculiarity of allowing global coverage with high vertical resolution. During the last thirty years several limb observation instruments have been launched in space. Examples of still operational satellite limb instruments are the Atmospheric Chemistry Experiment-Fourier Transform Spectrometer (ACE-FTS) (Bernath et al., 2005), that uses the solar occultation technique in the infrared, and the Microwave Limb Sounder (MLS) (Waters et al., 2006), that measures the atmospheric emission in the sub-millimeter region.





In March 2002 the European Space Agency (ESA) launched the ENVIronmental SATellite (ENVISAT), with on board three instruments capable to observe the Earth's atmosphere at the limb. Two of them (the SCanning Imaging Absorption spectroMeter for Atmospheric CartograpHYon - SCIAMACHY (Bovensmann et al., 1999) and the Global Ozone Monitoring by Occultation of Stars - GOMOS (Kyrölä et al., 2004)) operated in the Near IR, visible and UV spectral regions. The third one, the Michelson Interferometer for Passive Atmospheric Sounding (MIPAS) (Fischer et al., 2008), measured the infrared limb emission of the atmosphere in the 680-2410 $cm^{-1}$ spectral region. MIPAS scanned the atmospheric limb in the orbit plane, enabling a quite dense vertical and latitudinal coverage of the atmosphere (sampling steps smaller than 3 km in the vertical domain and smaller than 500 km in the horizontal domain), that has enabled to investigate both the vertical and horizontal atmospheric variability over the whole duration of the ENVISAT mission (2002–2012).

The Optimized Retrieval Model (ORM) algorithm (Ridolfi et al., 2000) was developed to be the scientific prototype of the ESA level 2 code, used to routinely analyse the middle-infrared limb emission measurements acquired by the MIPAS instrument. Over the years, the code has been improved both to follow the evolution of the measurement strategies and to adopt new strategies suggested by the analysis of the quality of the retrieved products (Raspollini et al., 2006, 2013). Besides ESA data, there are a series of independent level 2 databases produced with MIPAS observations, at Oxford University (Dudhia et al., 2007), at the Karlsruhe Institute of Technology (von Clarmann et al., 2009a, 2003; Milz et al., 2005; Kiefer et al., 2021), and at ISAC-CNR Bologna (Dinelli et al., 2010), which include different targets and use different analysis strategies. Recently, a new version of ORM, ORM v8.22 (Raspollini et al., 2021) has been developed for processing the final release of MIPAS level1b data, and produce the final version of the ESA level 2 MIPAS products. For the first time, the scientific version of ORM has been used to routinely process MIPAS measurements and produce the MIPAS level 2 version 8 (level2-v8) database. The version 8 of MIPAS level2 database includes data covering the complete MIPAS mission for both the main targets (pressure, temperature, and VMR of $H_2O$, $O_3$, $HNO_3$, $CH_4$, $N_2O$, $NO_2$), originally selected for the ESA near real time retrievals, and 15 additional minor species which, over the years, were found to be retrievable from individual limb scans. The MIPAS level2-v8 database, now available to the end users, covers the complete MIPAS mission including the measurements acquired in the original spectral resolution (0.035 $cm^{-1}$ from July 2002 to March 2004), and in its final configuration (from 2005 MIPAS spectral resolution has been set to about 0.0625 $cm^{-1}$) spanning a time lapse of ten years. The database (ESA - ENVISAT MIPAS L2, 2021) can be accessed from the ESA web site https://earth.esa.int/eogateway/. In this paper we will describe all the aspects of the database, from the auxiliary data to the retrieval choices adopted for each trace species. The products present in the MIPAS level2-v8 database will be characterised in terms of their retrieval error, spatial resolution, and 'useful' vertical range. The strategy used to identify good quality profiles will be also described along with an easy way to identify them.

## 2 The MIPAS instrument

MIPAS design was originally developed by developed by the Karlsruhe Institute for Technology (Fischer et al., 1996) and adopted by ESA as one of the core instruments of the ENVISAT satellite that was successfully launched on $1^{st}$ March 2002 and was continuously operated until the $8^{th}$ April 2012, when connections with the satellite were suddenly lost. ENVISAT was



on a near-circular sun-synchronous orbit at 800 km altitude (98.55 deg inclination) and, with an orbit period of 100.6 min, it performed 14.3 orbits per day, with a 10 a.m. descending node and a repeating cycle of 35 days. MIPAS was a Fourier Transform (FT) spectrometer recording the limb emission spectrum of the atmosphere in the mid-infrared (from 680 to 2410 $cm^{-1}$). The full spectral coverage of MIPAS was achieved through 5 spectral bands: Band A (680-980 $cm^{-1}$), Band AB (1010-1180 $cm^{-1}$),

Band B (1205-1510 $cm^{-1}$), Band C (1560-1760 $cm^{-1}$), and Band D (1810-2410 $cm^{-1}$). A more detailed description of the MIPAS experiment can be found in Fischer et al. (2008).

Most of MIPAS measurements were performed with the instrument Line Of Sight (LOS) pointing backward along the orbit track, a part from few measurements used for monitoring air-plane wakes and volcanic eruptions. For the measurements acquired over the polar regions the instrument azimuth was tilted to compensate for the orbit inclination and enable their full

coverage. Near the tangent point, MIPAS instantaneous field of view was about 3 km in altitude and 30 km across track. The first part of MIPAS measurements (which lasted until the $26^{th}$ of March 2004) were performed with a spectral resolution of 0.035 $cm^{-1}$ Full Width Half Maximum, unapodized. This part of the mission is called the Full Resolution (FR) phase. Most of FR MIPAS measurements were acquired in the nominal (NOM) observation mode, where each limb-scan was made of 17 limb views (sweeps) with nominal tangent altitudes ranging from 6 to 68 km with the average distance between the tangent points

of successive limb-scans of about 500 km, i.e., ∼4.7 degrees in latitude.

Due to technical problems with the interferometric slides, the FR mission was suspended in April 2004. MIPAS operations were resumed in January 2005, operating MIPAS at an Optimized Resolution (OR), reducing the spectral resolution to the 41% of the maximum spectral resolution of the original configuration. Since the reduced spectral resolution implied a lower acquisition time for each limb view, all the MIPAS observation modes were optimised in order to exploit at the maximum the

new instrumental feature. Therefore, in the OR nominal observation mode, each scan was made of 27 limb views with tangent altitudes ranging from 3 to 70 km with fixed altitude steps, and the average distance between the tangent points of successive limb-scans of about 400 km (∼3.7 latitude degrees). While the vertical spacing between the tangent altitudes of the spectra of the individual limb-scans was kept constant along the orbit, the whole scan pattern was shifted with latitude to follow the tropopause altitude. MIPAS was operated at 100% of its duty cycle in the FR mission. Because of the mentioned instrument

anomaly, the operations were reduced significantly from 2005 to 2007. At the beginning of 2005, MIPAS was operated at only a 30% duty cycle, which was progressively increased until December 2007, when it was successfully recovered back to 100% . After the $21^{th}$ October 2010, the ENVISAT platform was moved to a lower altitude with drifting orbit, but the observation modes and duty cycle of MIPAS remained unchanged until the unexpected loss of contact with the satellite ended the MIPAS mission on the $8^{th}$ of April 2012.

During its lifetime, MIPAS was operated in many observational modes. The most important and most frequent ones are the 'nominal' modes (NOM), with different latitude-altitude patterns in the FR and OR parts of the mission. In the FR NOM observation mode the nominal tangent altitudes were acquired from 6 to 42 km with 3 km steps, and at 47, 52, 60 and 68 km. In the OR NOM observation mode the nominal tangent altitudes were: 6.0, 7.5, 9.0, 10.5, 12.0, 13.5, 15.0, 16.5, 18.0, 19.5, 21.0, 23.0, 25.0, 27.0, 29.0, 31.0, 34.0, 37.0, 40.0, 43.0, 46.0, 50.0, 54.0, 58.0, 62.0, 66.0, and 70.0 km. During the FR mission, a

small fraction of MIPAS observations were acquired using two different special observation modes (S1, dedicated to polar



Chemistry on Feb 16, 2004, and S6, dedicated to measure the Upper Troposphere-Lower Stratosphere - UTLS- on Sept 19, 2003), designed to improve MIPAS coverage for special scientific objectives. These modes exploited different spectral resolutions, vertical scanning range and sampling steps, and could use sideways views. During the OR mission, several different observation modes were routinely used (De Laurentis, 2005; Oelhaf, 2008; Dudhia, 2008). The most frequent duty

cycle, repeated every 10 days, consisted of 4 consecutive days of NOM measurements, 1 day of observations in the Upper Atmosphere (UA) mode, where a wider altitude range at reduced vertical resolution was measured, 4 days of NOM mode, 1 day of Middle Atmosphere (MA) mode. Other used observation modes were the UTLS-1, and the NLC (Noctilucent Clouds) modes. All the observation modes different from the NOM ones are often referred to as 'special modes'. Table 1 reports all the observation modes analysed to produce the level2-v8 database and their vertical range, while Fig. 1 reports the location of the

lowest tangent altitudes as a function of the latitude for both FR and OR NOM and the UTLS-1 modes.

## 2.1 ESA MIPAS Level 1b data

The first of the two main processing steps of the MIPAS measurements produced ESA Level 1b data. Level 1b processing decoded the downlinked source packets of MIPAS interferograms transforming them into geolocated and calibrated atmospheric radiances (Level 1b spectra). The calibrated spectra acquired during each orbit have been collected into one file, with the

starting and ending scan acquired over the equator. In this section we report some of the main features of the level 1 data that may have some influence on the quality of the final Level 2 data. Level 1b data for the FR and OR mission have been routinely released by ESA and updated in successive improved processing. The current MIPAS database of Level 1b data has been produced with the version 8 of the ESA Instrument Processing Facility (IPF) (L1v8, Kleinert et al., 2018). L1v8 processing includes many major improvements with respect to the previous versions, namely:

– **Better radiometric accuracy** obtained with an improved time-dependent in-flight characterisation of the detector nonlinearity.

    – **Less discontinuities in the time series** due to the use of the daily gain measurements (in previous processing the gain function was updated only once per week).

    – **Better accuracy in the reported LOS tangent altitudes**. Smaller residual pointing errors are found in L1v8 data with

135        respect to previous versions (lower than 200 m at high altitudes, and than 400 m at low altitudes).

The Level 1b spectra are reported on a frequency grid of $0.025\,\mathrm{cm}^{-1}$ for the FR mission and of $0.0625\,\mathrm{cm}^{-1}$ for the OR mission. Further details on the version 8 of the level1b processor can be found in Kleinert et al. (2018) and on the quality of L1v8 data in ESA-EOPG-EBA-TN-1 issue 1.0 (2019).

## 3 MIPAS Level 2 processor

Up to the version 7 release, MIPAS Level 2 data have been processed directly by ESA with the level 2 processor based on the ORM theoretical baseline and implemented by an industrial contractor. The Level 2 data of MIPAS main targets, namely



pressure at tangent altitudes, temperature, and the VMRs of $H_2O$, $O_3$, $HNO_3$, $CH_4$, $N_2O$, and $NO_2$, were continuously retrieved and distributed by ESA in near real time from the nominal mode measurements for the FR phase (Raspollini et al., 2006). However, the number of molecules included in the many re-processing of MIPAS data has increased over the years. In the OR phase, the near real time processing was performed only starting from June 2010. The ORM retrieval strategy is the following: each target is retrieved on a set of small spectral regions called MicroWindows (MWs), selected in order to have the best retrieval accuracy of the final Level 2 products (see Dudhia et al., 2002a, and Sect. 4.2.2 below). When analysing a single limb-scan, ORM performs a sequential retrieval of all the targets, using the results of the previously retrieved targets in the following steps of the retrieval sequence; the order used in the sequence was designed to minimise the error propagation. MIPAS limb-scans are individually analysed, retrieving the atmospheric quantities at the geolocation of the tangent points. As a consequence, the retrieved profiles are calculated on an irregular vertical grid, and the horizontal resolution is linked to the separation of subsequent scans (see von Clarmann et al., 2009b).

ESA Level2-v8 data have been obtained, for the first time, by directly using the ORM code, version 8.22 (ORM-v8.22, Raspollini et al., 2021). ORM-v8.22 implements the Global-fit (Carlotti, 1988) algorithm and performs the retrieval using the regularizing Levenberg-Marquardt method (Levenberg, 1944; Marquardt, 1963). Both, the Error Consistency (EC, Ceccherini, 2005) and the Iterative Variable Strength (IVS, Ridolfi and Sgheri, 2009, 2011) a-posteriori regularisation methods are implemented and can be selected according to the retrieved target. ORM v8.22 implements many improvements with respect to the previous versions. Here we list the major ones:

- **Accounting for the horizontal inhomogeneities**. The horizontal variability of the atmosphere is modelled in the forward model (FM) internal to ORM-v8.22 with user supplied horizontal gradients for both temperature and trace gases.

- **Altitude dependent cloud screening**. The cloud detection algorithm uses a cloud index (CI, Spang et al., 2004), defined as the ratio between the mean radiance in two spectral intervals with different sensitivity to the cloud radiative effects. Spectra with CI smaller than the pre-defined thresholds are not included in the analysis. In the previous versions, a single CI threshold, altitude and latitude independent, was used. ORM-v8.22 uses altitude-dependent and latitude-dependent cloud filtering thresholds (Sembhi et al., 2012). This approach was proven to be more efficient in detecting high-altitude clouds.

- **Optimal Estimation**. The Optimal Estimation (OE) or maximum a posteriori approach (Rodgers, 2000) has been implemented in ORM-v8.22. This method is used in the retrieval of new species with a very weak signal in MIPAS spectra. The a-priori information can be either equal to the initial guess profile or provided independently.

- **Initial Guess strategy**. The choice of the initial guess profile and of the assumed profiles of the interfering species has changed in the latest version of ORM. As described in Raspollini et al. (2021), different databases are available for those profiles and for the computation of the gradients. The selection of the profiles among all the available ones is made according to a user-defined priority list.

For further details of the new features implemented in ORM-v8.22 see the paper by Raspollini et al. (2021).



## 4 MIPAS level2-v8 products development strategy

In this Section we summarise the strategy adopted in the development of the MIPAS level2-v8 dataset, describing the retrieval setup and the auxiliary data used for the analyses of the whole MIPAS mission.

### 4.1 Retrieval strategy

All the data in the level2-v8 dataset have been obtained with ORM v8.22, configured as described in this section. As already reported in Sect. 3, ORM v8.22 performs a sequential retrieval of all targets at the tangent altitudes of the measured scan. The first step of the retrieval chain is the retrieval of the pressure at the tangent points jointly to the associated temperatures ($p, T$ retrieval). Then, the VMRs of the minor gases are retrieved, starting from water vapour. The adopted sequence of the VMR retrieval is: first the main target species $H_2O$, $O_3$, $HNO_3$, $CH_4$, $N_2O$, and $NO_2$, then the additional species $CFC-11$, $ClONO_2$, $N_2O_5$, $CFC-12$, $COF_2$, $CCl_4$, $HCN$, $CF_4$, $HCFC-22$, $C_2H_2$, $CH_3Cl$, $COCl_2$, $C_2H_6$, $OCS$, $HDO$ are retrieved in the reported order. In the retrieval sequence the initial VMR profiles, used for the $p, T$ retrieval, are replaced by the ones already retrieved, with the exception of the ones of some species which have a very weak contribution in MIPAS spectra. Therefore, the retrieval sequence and the used spectral intervals were designed to minimise the impact of the systematic errors originated by the non-correct modelling of the interfering species. Each target is retrieved simultaneously with the transmission profiles of the atmospheric continuum (one for each MW) in the altitude range of 6-25 km for the $p, T$ retrieval, and of 6-30 km for all the other targets, along with an altitude and frequency independent radiance offset for each analysed MW. The majority of the retrievals are performed using the Levenberg-Marquardt technique (Levenberg, 1944; Marquardt, 1963), in order to cope with forward model non linearities, and an a-posteriori regularisation to avoid unphysical oscillations in the final results. For the molecules $HCN$, $CF_4$, $C_2H_2$, $C_2H_6$, $CH_3Cl$, $OCS$, $COCl_2$, and $HDO$, all characterised by very weak signals in MIPAS spectra, the OE technique has been used.

### 4.2 Auxiliary data

#### 4.2.1 Initial guess and a-priori data

The atmospheric state, used both as a starting point in the retrieval iterations and to represent the vertical distribution of the non-retrieved interfering species, is represented with vertical profiles of pressure, temperature and VMR of all the gases included in the simulation of the measured spectra with the forward model internal to the retrieval code. In ORM V8.22, the atmospheric state includes also the horizontal gradients introduced to model the atmospheric horizontal (latitudinal) inhomogeneity. Obviously, part of the atmospheric state are invariable profiles (background), which remain fixed during the retrieval iterations, and variable profiles, which are part of the retrieval vector and are modified during the retrieval process. The strategy for the choice of the initial and background atmospheric state has been revised in ORM v8.22, increasing its flexibility but retaining the underlying rationale that has been successfully applied in the previous versions. As reported in Raspollini et al. (2021) the initial atmospheric status can be built using different sources that can be chosen according to their





availability and their priority. The standard choice for the development of MIPAS level2-v8 database is to use, in order of priority, the results from the already completed retrievals of the same scan, the results from retrievals of the previous scan and the profiles extracted from the IG2 database, developed by (Remedios et al., 2007) and recently updated as described in (Raspollini et al., 2021). This replicates the behaviour of the previous versions of the level 2 code. For the gradients (used only

for temperature, $H_2O$ and $O_3$) vertical profiles extracted from the ECMWF ERA Interim reanalysis at the geolocation and time of each scan are used. Since the vertical range of the used profiles may not cover the full vertical range used by the forward model, to extend them the corresponding IG2 profile, scaled in order to maintain the continuity with the original one, is used. Gradients are extended with 0 values outside the covered vertical range.

The OE method is only used for the retrieval of the species with a very weak signal in MIPAS spectra, listed in table

2. For all these targets, except HDO, the a-priori information is fixed for all the analysed scans (one vertical distribution per target, provided as a function of pressure, for all latitudes and seasons). For HDO, the used a-priori information is the vertical distribution of $H_2O$ retrieved from the same scan, scaled by the natural $HDO/H_2O$ isotopic ratio. The fixed a-priori profiles have been obtained averaging the IG2 climatological profiles at fixed pressure levels. The use of a fixed a-priori profile guarantees that the observed variability (estimated as a function of pressure) in the retrieved products represents the variability

observed by MIPAS. In the OE inversion procedure the full a-priori co-variance matrix (CM) is used. This matrix is internally built by ORM-v8.22 using as diagonal elements the squares of the a-priori errors, computed as a fixed percentage of the a-priori values plus a constant term introduced to prevent strong constraints in case of very small a priori data. The off-diagonal terms are then built introducing a correlation that decays exponentially with the vertical distance from the considered retrieval altitude, and its strength is tuned via a correlation length that is target-dependent. Table 2 reports the values of the a-priori errors and

correlation lengths used for all the targets for which OE is used.

### 4.2.2 Microwindows and look-up tables

As already introduced in section 3 and reported in Ridolfi et al. (2000), the MWs are the small regions of the spectrum selected for the retrieval of any particular species. For MIPAS, a maximum MW width of 3 cm$^{-1}$ is imposed, originally for reasons of computer memory. This corresponds to 121 spectral points at the original FR spectral sampling and 49 for OR. MWs are

generally also used over limited tangent altitude ranges, so a MW can be considered a rectangle in the total measurement domain represented by the collection of spectra within a complete limb scan.

Intuitively, one might expect MWs to include all the prominent lines of the target molecule ($CO_2$ in the case of the $p, T$ retrieval), together with a portion of the background from which the continuum level can be established and, prior to MIPAS, MWs were often selected subjectively with these aims. For MIPAS, however, an automated MW selection algorithm was

developed (Dudhia et al., 2002a) which aimed not only to maximise the information on the target species, but also to minimise the contributions of systematic errors such as interfering species and non-Local Thermodynamic Equilibrium (LTE) effects. This is achieved by the selection algorithm creating 'spectral masks' – the deliberate masking out of spectral points at a particular tangent altitude whose inclusion in the operational retrieval would introduce, in terms of overall information content, an increase in the systematic error larger than the obtained reductions in the noise error. The MW selection algorithm provides





not only the MWs to be used in the retrieval but also the altitude range where they are applied. This information is included in a file, called 'Occupation Matrix', and it is used by the Level 2 algorithm to extract the portion of the spectra to be analysed for each tangent altitude. The selection tool is also used to estimate the residual systematic errors that may affect the level 2 products obtained with the selected set of MWs (see sect. 5.2.2). Tables **??** through **??** in the supporting material report the list, the frequency regions and the altitude coverages of all the MWs used to develop MIPAS level2-v8 database, divided by species

and observation mode for which they have been used.

The forward model within the ORM retrieval algorithm is based on radiative transfer calculations at 0.0005 cm$^{-1}$ frequency grid, set by the requirement to capture the Doppler cores of high altitude spectral lines. This typically leads to around 7500 points required to model a single MW at a single tangent height (the Instrumental Line Shape convolution requires modelling beyond the MW boundaries). A line-by-line calculation was considered too slow (at least at the start of the mission) so look-up

tables of the absorption coefficient $k$, tabulated as a function of wavenumber, pressure and temperature, were used; one for each molecule with significant absorption features within each MW. Approximately 1000 $p, T$ tabulation points are required to avoid large interpolation errors, so such tables containing about $10^7$ data points are unwieldy; so further compression was obtained using singular-value decomposition. Further details can be found in (Dudhia et al., 2002b).

For the development of the MIPAS level2-v8 database, as described in Raspollini et al. (2021), the look-up tables have been

computed using the dedicated spectroscopic database (MIPAS spectroscopic database version pf4.5), complemented by the HITRAN 2016 (Gordon et al., 2017) tabulated cross sections for all the heavy molecules except CFC$-$11 (Harrison, 2018), CFC$-$113 (Le Bris et al., 2011) and SF$_6$ (Manceron et al., private communication).

## 5  MIPAS level2-v8 products characterisation

The MIPAS level2-v8 database, along with the values of tangent pressures, temperatures and VMR profiles of all the retrieved

molecules, includes also some important products that can be used as diagnostic tools to characterise the quality of the reported results. Among them, the averaging kernels, the covariance matrices that map the random measurement noise onto the solution, and a few quality flags. All the products are stored in netcdf files. The following subsections describe some of the characterisation quantities and the results of the quality control of MIPAS level2-v8 products while the description of the format and the content of the output netcdf files is described in appendix A.

### 5.1  Averaging Kernel Matrix

The elements of the averaging kernel matrix (AKM) represent the sensitivity of the retrieved profile to the real atmospheric vertical distribution. The AKM enables to assess the amount of information provided by the observations accounting for the constraints used in the retrieval (i.e. the applied regularisation or the a priori information). The AKM for each retrieved profile is computed during the retrieval procedure. For the species retrieved using the Levenberg-Marquardt regularising method, the

AKM is calculated taking into account all the steps performed during the retrieval iterations, as described in Ceccherini and Ridolfi (2010). When the a posteriori Tikhonov regularisation is applied, the AKM is updated accordingly (Ridolfi and Sgheri,





2011). For the species retrieved using OE the AKM is evaluated with the standard procedure (Rodgers, 2000). In all cases the AKM is computed on the vertical retrieval grid of the considered scan. Figure 2 reports an example of the AKs computed for scan 70 of orbit 36275: the left panel shows the AKs for ozone, a species retrieved with the Levenberg-Marquardt method,

and the right panel show the AKs for HCN, a species retrieved with OE. We see in the figure that while the peaks of the AKs for Ozone are always close to 1 (due to the weak constraints imposed by the a-posteriori regularisation), the peaks of the AKs for HCN are always below 0.8. We also see a larger width of the AKs of HCN, due to the off diagonal terms used in the construction of the a-priori CM that correlates nearby retrieved points.

The AKM of each retrieved profile is included in the database files, as described in appendix A, and is used to compute the

number of degrees of freedom (DOF) of each profile and other quantifiers linked to the quality of the dataset products (see Sect. 5.3.2). Table 3 reports the average number of DOFs of all the species for the scans unaffected by clouds in both the FR and OR measurements in the nominal mode, along with the number of retrieved points for all the species included in the database. We see in the table 3 that the number of tangent points used in the retrieval is on average larger in the OR phase. This is due to the finer vertical measurement grid adopted in the OR phase of the MIPAS mission. We also see that the number of DOFs

for temperature, water vapour, ozone and nitric acid is significantly larger in the OR phase than in the FR one, while for other gases they remain unaltered or are slightly smaller. This can be explained by the combined effect of the finer vertical sampling in the OR phase obtained at the expenses of a coarser spectral sampling (and resolution) that affects in a different way the retrieved species.

## 5.2   Accuracy and precision

Precision and accuracy of the retrieved profiles are represented by the random and the systematic components of their errors. What we call the noise retrieval error represents the mapping of the noise equivalent spectral radiance of the measurements onto the retrieved parameters. This mapping is quantified by its covariance matrix (CM), computed during the retrieval procedure. Among the systematic errors, the forward model errors are caused by the approximations used in representing the radiative processes of the atmosphere and the instrumental effects in the retrieval process (Dudhia et al., 2002a). Some of the forward

model errors are random, like the propagation of the noise errors affecting the retrieved temperature and pressure to the retrieved VMR profiles ($p, T$ propagation error, see Raspollini and Ridolfi, 2000), and some are systematic, such as the spectroscopic errors. Others have values that may change with time or depend on the spatial scale of the profiles that are considered for the statistical analysis.

### 5.2.1   Random errors

The main contributions to the random error component are the noise error and the propagation of the random error of the retrieved pressure and temperature in the subsequent VMR retrievals ($p, T$ propagation error). The noise error is estimated in the retrieval procedure by performing the square root of the diagonal elements of the CM computed at the end of the retrieval iterations. Similarly to the approach used for the AKM calculation (see Sect. 5.1), the computation of the CM takes into account both the Levenberg-Marquardt iterations and the performed regularisation, or, for the weak species, the used a-priori





information. In the MIPAS level2-v8 database, the final CM is provided along with the squared root of its diagonal elements (the noise error, see Sect. A) for each scan.

The $p, T$ propagation error in the VMR retrievals is computed a posteriori using the CM of the retrieved pressure and temperature profiles of the same scan and pre-computed matrices obtained with test retrievals where temperatures and pressures at each level were perturbed (Raspollini and Ridolfi, 2000). In the database, the $p, T$ propagation error is provided for each

retrieved profile, see appendix A. The final random error for the VMR retrievals can be estimated by the square root of the sum of the squares of the single scan noise error and the single scan $p, T$ propagation error (i.e. the square root of the sum of the diagonal elements of the retrieval CM and the $p, T$ propagation error matrix).

In order to evaluate how the random errors change in different atmospheric conditions, the average of the single scan random errors has been computed for the five reference atmospheres, namely polar summer daytime, polar winter nighttime,

mid-latitudes (both daytime and nighttime) and equatorial atmospheres. Figure 3 for the Full Resolution case and Fig. 4 for the Optimised Resolution case show the average Ozone VMR profiles (left plots) and their associated average random error profiles, in absolute (middle plots) and relative (right plots) scale, averaged for the nominal mode only of the FR and OR phases of the mission. In can be seen in the figures that the contribution coming from the $p, T$ propagation error is generally smaller than the noise contribution, but it is not negligible, especially for the Optimised Resolution case.

### 5.2.2 Systematic Errors

Systematic errors are broadly all known sources of error in the retrieval apart from the random errors. These can be grouped into the following categories:

- **Approximations**: omissions in the physics of the radiative transfer model such as non-Local Thermal Equilibrium effects, $CO_2$ line-coupling, and use of look-up tables.

- **Atmospheric Errors**: uncertainties in the assumed profiles used to model the atmosphere.

- **Instrument Errors**: uncertainties in modelling the instrument, such as radiometric calibration or instrument line shape.

- **Spectroscopic Errors**: uncertainties in the spectroscopic data, i.e. line strengths and air broadening coefficients.

The systematic errors due to the above mentioned sources are calculated off-line by the MWs selection algorithm using linear error propagation. Each error source is expressed in the measurement space as a $1\sigma$ perturbation on the nominal spectral

values $\delta \mathbf{y}$ (for example, the expected change in the measured spectra if the profile of an interfering species is perturbed by its $1\sigma$ uncertainty). Then, using a model of the retrieval gain matrix $\mathbf{G}$, the error is mapped into the retrieved profile space as

$$\delta \mathbf{x} = \mathbf{G} \delta \mathbf{y}, \tag{1}$$

where $\delta \mathbf{x}$ is the estimated error on the profile due to this particular error source.

Since the pressure-temperature retrieval is fundamental to all other species, systematic errors propagated from the $p, T$

retrieval are formally combined with the direct systematic errors of each species. The systematic error budget for MIPAS


retrievals has been calculated off line for a set of 5 representative atmospheres, the same used to represent the $p, T$ propagation errors (see sect. 5.2.1), and it refers to the single scan profile. The systematic errors for the reference atmospheres are available at http://www.atm.ox.ac.uk/group/mipas/err/ and their value along with further details on their derivation and usage will be reported by Dudhia et al. (2021).

## 5.3 Filtering and quality flagging

The quality of the retrieved profiles is determined on the basis of four criteria, two providing information on the successful convergence of the retrieval iterations, one on the capability of the retrieval to reproduce the measurements and one on the presence of outliers in the retrieval error. To provide an easy way to remove unreliable data, a final post quality flag, summarising the outcome of the four quality criteria, is reported in the output files. The successful convergence of the retrieval iterations is obtained when the convergence criteria described in Raspollini et al. (2021) are met and the Marquardt parameter is sufficiently small not to trigger false convergence. The capability of the retrieval to reproduce the measurements is verified when the final $\chi^2$ is smaller than a given threshold, determined a posteriori from the distribution of the $\chi^2$ obtained in the retrievals of each target (see Raspollini et al., 2021). The presence of outliers in the retrieval errors has been found to be correlated to outliers in the retrieved profiles. Hence, only the profiles with maximum error smaller than a given threshold, obtained again a posteriori examining the maximum error obtained in a representative number of orbits, can be considered reliable. The post quality flag is an integer variable set to zero only when the four quality criteria are met. If one (or more) of the quality criteria is not met, the post quality flag is set to 1 and the whole profile should be discarded. The MIPAS level2-v8 database contains all the retrieved profiles of temperature, regardless their quality indicators, but only the VMR profiles of the trace species which have been measured in a scan where the post quality flag of the temperature profile is zero. Therefore, the users should always check the value of the post-quality flag of each reported profile.

Figure 5 shows the total number of Level 2 products (i.e. the number of profiles) available in the Level2-v8 repository for each processed trace species. The maximum number of products present in the repository is for the temperature retrieval, i.e. 2,993,597 profiles. When a temperature profile retrieved from a scan is not considered of good quality no other retrievals are performed, and this explains why the number of products present for all the other species is lower or equal to the number of accepted T profiles. About 2.94 millions of retrieved profiles are obtained for $H_2O$, $O_3$, $CH_4$, $N_2O$, $NO_2$, retrieved from all the analysed observation modes. About 2.74 millions are obtained for $HNO_3$, CFC-11, $ClONO_2$, $N_2O_5$, CFC-12 retrieved from Nominal, UTLS (including UTLS-1, UTLS-1-old and UTLS-2), Middle Atmosphere and AE only; 2.54 millions are obtained for $COF_2$, $CCl_4$, $HCN$, $CF_4$, $CFC-22$, $C_2H_2$, $CH_3Cl$, $COCl_2$, $C_2H_6$, $OCS$, $HDO$, retrieved from Nominal, UTLS and AE observation modes only.

Figure 6 points out the effect of the quality filtering on the total Level 2 products. For most molecules ($H_2O$, $O_3$, $CH_4$, $N_2O$, $NO_2$, $HNO_3$, CFC-11 $ClONO_2$, $N_2O_5$, CFC-12, $COF_2$, $CCl_4$, CFC-22, $CH_3Cl$, $COCl_2$) the fraction of good products ranges from 96 to 99% of the total number of profiles present in the output files; for other molecules, whose signal is very weak ($HCN$, $CF_4$, $C_2H_2$, $C_2H_6$, $OCS$, $HDO$) the fraction of good profiles is smaller.



MIPAS level2-v8 users should be aware that negative values have not been removed from the database in order to avoid
the introduction of biases when averaging the data. Moreover the data have not been filtered against the number of degrees of
freedom or strong deviations from climatological expectations to prevent the loss of information in case of exceptional events
in the atmosphere. Therefore, when handling the data, users should make use of the following recommendations.

### 5.3.1   Use of negative values

While negative VMRs are just retrieval artefacts, and therefore should be filtered out for the use of individual measurements,
they make perfect sense for statistical applications, given the non-zero uncertainty of the measurements and the retrieval
process. It is therefore important that negative values are not masked when performing averages. Masking out the negative
values could introduce a positive bias when the data are averaged as part of scientific analyses, especially for small VMR
values.

### 5.3.2   Useful vertical retrieval range

Each retrieved profile is properly and fully characterised on the full retrieval range provided in the output files by the corre-
sponding CM and AKM. Altitude regions with poor information on the retrieved target can be identified by the low values of
the diagonal elements of the AKM and/or the large values of the diagonal elements of the CM. Since the AKM and the CM
are calculated considering the retrieval on the full vertical range, the use of only a part of the profile, with the association of
sub-matrices of the provided AKM and CM, implies an approximation in the AKM and CM. Therefore it is recommended to
use the profile, with its CM and AKM, on the full vertical range.

However, to evaluate the altitude range where on average the retrieved values are 'useful', considerations on the physical
meanings of the measurements and on the degrees of freedom per unit height (DOF/height) distribution (Ceccherini et al.,
2013) have been used. The DOF/height for each retrieved point is computed as the ratio between the relative diagonal element
of the Averaging Kernel Matrix and the average step between the altitude of the considered point and the altitudes of the two
adjacent ones; its value is proportional to the information content of the observations at the corresponding tangent altitude. As
an example, Fig. 7 reports two examples of vertical distributions of the DOF/height for the same species used in Fig. 2.

For the species retrieved using the OE technique, the useful vertical range is identified as the range where the DOF/height
distribution assume values larger than a threshold of $0.05 \text{ km}^{-1}$, which corresponds, in an ideal condition of a triangular AK,
to a vertical resolution of 20 km. This criterium is not sufficient to identify an useful range for the retrievals performed using
the Levenberg-Marquardt technique alone, for which the profiles of some retrieved species, at either the high or low boundary
of the retrieval range, may be characterised by very small VMRs and larger retrieval error in correspondence to DOF/height
values larger than the threshold. In some of these cases, also negative values in the average profiles are found, indicating the
presence of a systematic error. Therefore, the altitude range where the averaged profiles are negative are excluded from the
useful range. Figure 8 reports, for each species, the nominal retrieval range for both FR (in blue) and OR (in red) parts of the
MIPAS mission, with superimposed the useful retrieval range, indicated by the shaded area. Note that the reported range for
the OR measurements, characterised by an altitude range variable with latitude, is relative to the middle latitudes. We stress


again that the information on the useful range must be used independently from the AKM and CM, that should be used taking into account the full vertical range of each product. This means that in the comparison of MIPAS data with other independent measurements, the MIPAS AKMs should be used in their full range while the comparison of the data themselves should be
evaluated only in the useful range.

## 5.4 Altitude grid representation

MIPAS measurements contain information on the pressures corresponding to the tangent altitudes of the scan. The Level 1b files report for each limb view the so called 'engineering tangent altitudes', the altitudes of each tangent point estimated from the information on the platform position and the instrument commanded pointing. In the Level 2 processing, for every measured
scan, the pressure and temperature corresponding to each tangent point are retrieved. Then, the altitude grid is rebuilt from the retrieved pressure and temperature profiles using the hydrostatic equilibrium equation and assuming the lowest engineering tangent altitude as correctly known. Any error in this anchor point will lead to an artificial shift of the entire altitude grid. For the Level 2 data versions prior to V7, the error on the anchor point was considerable and it was therefore recommended to use only pressure as the vertical coordinate for the retrieved profiles.

From version 7 onward, a large effort on the Level 1 processor development has led to a more accurate estimate of the engineering tangent altitudes (Kleinert et al., 2018) by using a model to correct for both the seasonal cycle and the negative trend in the differences between engineering altitudes and correlative measurements. The Level 2 processor (starting from version 7), on the other hand, corrects the lowest engineering tangent altitude using information from co-located ECMWF altitude/pressure profiles, when available (Raspollini et al., 2013). This considerably reduces the error on all the altitude range. Therefore, the
data included in the MIPAS level2-v8 database can be represented as a function of pressure or altitude with the same accuracy. MIPAS level2-v8 altitude versus pressure profiles were compared to coincident in-situ radiosondes measurements collected across the whole globe during the entire mission. The results of this validation are reported in the document ESA-EOPG-EBA-TN-1 issue 1.0 (2019). The median bias between MIPAS and the reference data is lower than 20 m at the lowermost altitudes (anchor points) and gradually increases to reach the largest value of 100 m at 10 hPa. A similar vertical dependence is also
noted in the spread of the comparisons, with values of 40-50 m at the anchor point, gradually increasing to 100-150 m around 10 hPa. The latitudinal dependence of these quality indicators is generally negligible.

## 6 The MIPAS level2-v8 database

As reported in the previous sections, the MIPAS level2-v8 database has been obtained with the latest (and best) MIPAS level 1b data, and with the most updated version of the ORM code. Many shortcomings of the previous versions have been corrected
and new species have been added. The current release includes the profiles at tangent pressures of the atmospheric temperature and of the VMRs of the following species: $H_2O$, $O_3$, $HNO_3$, $CH_4$, $N_2O$, $NO_2$, $CFC-11$, $ClONO_2$, $N_2O_5$, $CFC-12$, $COF_2$, $CCl_4$, $HCN$, $CF_4$, $HCFC-22$, $HDO$, $C_2H_2$, $C_2H_6$, $CH_3Cl$, $OCS$ and $COCl_2$.





The full database has a doi number (10.5270/EN1-c8hgqx4) and can be downloaded (the whole set or part of it) from the web site https://earth.esa.int/eogateway/, after registration (see ESA - ENVISAT MIPAS L2, 2021). The web site contains
more detailed technical instructions for the users and will be updated in case of new releases. In total, 35060 orbits have been successfully analysed, with 440 not processed because of problems with the level1b files. The statistics of the successfully retrieved vertical profiles in the case of temperature is reported in Fig. 9. In the maps it can be seen that the number of measurements change with pressure and with latitude band. This is due to the fact that the altitude (pressure) range over which the retrievals are performed changes from scan to scan due to cloud coverage.

To show the potentiality of the MIPAS level2-v8 data we report here a few examples. Figure 10 shows the time series of the vertical distribution of temperature, obtained performing monthly means of the retrieved temperatures over 69 pressure bins in six latitude bands, namely Polar North (covering the latitudes from $65°N$ to $90°N$), Mid Latitude North (covering latitudes from $30°N$ to $65°N$), Equatorial North (covering latitudes from $0°$ to $30°N$), Equatorial South (covering latitudes from $0°$ to $30°S$), Mid Latitude South (covering latitudes from $30°S$ to $65°S$), and Polar South (covering the latitudes from $65°S$ to $90°S$).
No distinction among day and night measurements has been made. The colour scale is reported in the figure and is the same for all the panels on the same vertical, to easily compare the results. The white vertical bands show the periods where MIPAS measurements are not available due to the instrumental problems reported in Sect. 2. The panels on the left hand side of Fig. 10 clearly show that the seasonal variability of the temperature is more enhanced in the South polar region than in the North, while comparable variability can be seen in both hemispheres for the equatorial and mid latitudinal bands. While the absolute values
of the temperature does not show remarkable differences between the FR and OR parts of the mission, the average random errors are consistently smaller in the FR period. This is mainly due to the larger number of spectral points (10 MWs used in FR while only 5 MWs used in OR) used to retrieve a smaller number of temperature points (at 17 tangent altitudes in FR instead of 27 in OR), and to the different vertical resolution (higher in the OR phase) of the products in the two parts of the mission.

Similar plots are shown for the water vapour results. Fig. 11 shows the time series of the vertical distribution of the monthly
means of water vapour as a function of pressure and for the same latitude bands of Fig. 10. The right hand panels clearly show that water vapor in the stratosphere is retrieved with average random errors always below 10%. As for the temperature, the FR part of the mission shows on average random errors lower than the OR part, despite the similar number of MWs used in the analysis (6 for FR and 5 for OR). This is again due to the larger number of spectral points used in the analyses of the FR measurements and to the different vertical resolution of the retrieved profiles in the two parts of the mission. The panels relative
to the equatorial regions clearly show the tape recorder effect of the water vapour distribution.

A couple of examples of the atmospheric polar phenomena that MIPAS has enabled to reveal are the identification of the polar vortex split-up in Antarctica in 2002 (Simmons et al., 2003; Glatthor et al., 2005), and one of the few observations of the ozone depletion in the North pole (Arnone et al., 2011). The two events are shown in Figs. 12 and 13.

Figure 14 shows the time series of ozone and $HNO_3$ in the Southern polar region along with all the chlorinated species
measured by MIPAS in the altitude region from 100 to 0.1 hPa. The seasonal cycle of those species, all involved in the ozone chemistry, is clearly visible. We see a clear indication of the seasonality in the formation of $ClONO_2$ at a pressure level of 2-3





hPa and its transport to lower altitudes, with a peak at 20-30 hPa, in coincidence with the start of the ozone depletion. All the other chlorinated species have a seasonal behaviour, with $CH_3Cl$ showing a six month cycle in the 20-30 hPa region.

Figure 15 shows the time series of the minor species, most of them retrieved for the first time in the frame of the ESA

processing of MIPAS data, in the mid-latitude Northern region. The data have been plotted only in the 'useful' vertical range reported in Fig. 8. For some molecules (i.e. HCN or $CH_3Cl$) a discontinuity in their vertical behaviour between the FR and OR parts of the mission can be noticed. This behaviour is observed in many trace species of the different versions of MIPAS dataset and it has to be taken into account when performing trend analyses, as shown in Valeri et al. (2018) for $CCl_4$. This is in part due to the different spectral microwindows selected for the retrieval in the two phases, but mainly it is due to the different

vertical sampling that makes the vertical resolution of the retrieved products different.

Figure 16 shows the time series of the latitudinal distribution at the pressure level close to the peak of the DOF/height curve for some weak species. The pressure level used for each species is reported in the title of each panel. For $CFC-11$ and $CFC-12$ a reduction of their VMR with time at all latitudes is clearly visible, while for $CFC-22$ we clearly see an enhancement of its concentration with time, as already reported by Chirkov at al. (2016). No clear behaviour is visible for

$CFC-14$ while we see a weak reduction with time for both $COF_2$ and $CCl_4$. HCN shows an asymmetric behaviour in the Northern and Southern hemispheres, possibly due to the occurrence of fires in Africa and in the Amazonian forest, that are both located in the Southern hemisphere. A similar asymmetric behaviour is visible in the panels relative to the species $C_2H_2$ and $C_2H_6$ where it can clearly be seen that their concentrations are larger in the Northern hemisphere. Finally, the seasonality of the concentrations of all these molecules is clearly visible in these maps.

**7   Conclusions**

The observations acquired during the full mission of the MIPAS instrument, on board the ESA ENVISAT satellite, have been analysed with the version 8.22 of the retrieval code ORM, originally developed as the scientific prototype of the ESA level 2 processor for MIPAS observations, to obtain the MIPAS level2-v8 database containing atmospheric fields of pressure, temperature and VMR of MIPAS main targets $H_2O$, $O_3$, $HNO_3$, $CH_4$, $N_2O$, and $NO_2$, along with the additional targets

$CFC-11$, $ClONO_2$, $N_2O_5$, $CFC-12$, $COF_2$, $CCl_4$, $CF_4$, $HCFC-22$, $C_2H_2$, $CH_3Cl$, $COCl_2$, $C_2H_6$, OCS, and HDO. The database covers the measurements acquired by MIPAS in most measurements modes both in the FR part (from July 2002 to March 2004) and the OR part (from January 2005 to April 2012) of the mission. We have described the auxiliary data and the retrieval choices used in the development of the MIPAS level2-v8 database. Each product is characterised in terms of the retrieval error, spatial resolution, and 'useful' vertical range in both phases of the MIPAS mission. We have shown that

these depend on the characteristics of the measurements (spectral and vertical resolution of the measurements), on the retrieval choices (number of spectral points included in the analyses, number of altitudes included in the vertical retrieval grid), and on the information content of the measurements for each trace species. For temperature, water vapour, ozone and nitric acid the number of degrees of freedom is significantly larger in the OR phase than in the FR one, mainly due to the finer vertical measurement grid. In the FR phase some trace species are characterised by a smaller retrieval error with respect to the OR



phase, mainly due to the larger number of spectral points used in the analyses, along with the reduced vertical resolution. We have discussed how possible caveats (negative VMR, vertical grid representation) can be handled. We have shown how the quality of the retrieved profiles is assessed through four criteria, two providing information on the successful convergence of the retrieval iterations, one on the capability of the retrieval to reproduce the measurements, and one on the presence of outliers. An easy way to identify and filter the problematic profiles with the information contained in the output files has been provided.

The number of species included in the MIPAS level2-v8 dataset makes it of particular importance for the studies of stratospheric chemistry. The span of time covered by the database, and the accurate calibration of MIPAS measurements, enables studies of trends for the stratospheric and upper tropospheric concentration of the included gases. The data are reported at the vertical grid coincident with the tangent altitudes of the observations, spanning the altitude range from 6 to 68 km in the FR and from 3 to 70 km in the OR. In the latitudinal domain, FR profiles are spaced by about 4.7 latitudinal-degrees while

the OR profiles are spaced by about 3.7 latitudinal-degrees. MIPAS level2-v8 data are available to the scientific community through the ESA Earth Observation portal (ESA - ENVISAT MIPAS L2, 2021).

## Appendix A:  MIPAS level2-v8 database file format

A big effort has been made to simplify the accessibility to the MIPAS level2-v8 data with respect to the previous versions. The level2-v8 database is distributed in netCDF-4 format (Rew et al., 2021), a format commonly used for the distribution of

the products of many ESA and NASA missions, and supported by a wide range of data-analysis packages and programming languages. With respect to previous versions of the level2 output files, the content of the MIPAS level2-v8 files has been designed to simplify the use of the data, without unnecessary duplication of information, and new variables have been added to ease the characterisation of the data.

     The information has been divided into two types of files, a standard one and an extended one. The names of both files are a

string of common format, and includes the name of the species, the measurement start and stop time, the orbit number, and a two-digits number identifying the file version: *MIPAS_[filetype]_[species]_[Start_time]_[Stop_time]_[orbit#]_[version].nc*

     The standard files, one for each orbit and retrieved species, contain the information commonly required by the data users. Its *filetype* label is '2PS', and it is compliant with the Climate and Forecast convention (CF-1.6, Eaton et al., 2011) and with the Attribute Convention for Data Discovery (ACDD-1.3, ESIP, 2015). The file contains, for each orbit's scan, its sequential

number, as reported in the related Level1b file, which allows a one-to-one correspondence between Level1b and Level2 files, and a flag that identifies the observation mode used for the measurement of that scan. Along with the results obtained for the target species (temperature or VMR with their corresponding errors) the information included are: the time, the scan geolocation, the sun illumination condition (day/night), the quality control variables (chi-square and cost function), final value of the Marquardt parameter, the convergence flag, the quality flags, the pressure and altitude grids, the cloud index (Raspollini

et al., 2021) (reported only for the temperature files), the retrieved temperature and related errors (repeated in all the VMR files), the blocks of the full covariance matrix and vertical averaging kernel matrix relative to the profile of the target species only, the a priori information (only if the trace species has been retrieved with the OE approach), and the co-variance matrix due

to the propagation of the pressure and temperature retrieval error on the VMR trace species retrieval ($p, T$ propagation error). In the standard files, the vertical grid over which all the retrieved data are reported is common to all the retrieved species of the

considered limb scan and corresponds to the tangent pressure grid of the limb sequence used for the pressure and temperature retrieval. The profiles values of each species at the altitude not included in the retrieval grid are reported with a dummy field (-88888.8). The size of the vectors and matrices are the same for all the analysed observation modes and it is equal to the largest number of tangent altitudes included in a limb sequence (27). When the analysed observation mode implied the measurement over a lower number of altitudes, the otherwise empty fields are filled with a -99999.9 number. This strategy enables the user

to distinguish a missing altitude because of filtering criteria or retrieval choices from a missing altitude due to the considered observation mode.

Extended files, identified by the *filetype* label '2PE', are also provided for each species and each orbit. They are thought for diagnostics and for advanced users, who need complete information about the retrieval process. It includes the full state vector (retrieved profiles, atmospheric continuum and instrumental offset) along with the full CM and AKM, and additional

information about the retrieval. In case of pressure and temperature retrieval, both profiles are included in the state vector. Because of the variables included into the full state vector, the retrieved parameters in the extended file are reported with different units. As a consequence, the extended file cannot be compliant with the Climate and Forecast convention and with the Attribute Convention for Data Discovery. Further and detailed information can be found in the technical note describing the format and contents of the files (Barbara and Raspollini, 2020) that can be found in the same site where the MIPAS level2-v8

database is stored. The table with the complete list of the variables included in the two types of files is reported in table S32 of the supporting material.

*Data availability.* The data are available at the following web site
https://earth.esa.int/eogateway/catalog/envisat-mipas-temperature-pressure-and-atmospheric-constituents-profiles-mip_nl__2p-

*Author contributions.* All co-authors contributed to the development of the database through the participation to MIPAS Quality Working

Group. AD and MLP worked mainly to the microwindows, LUTs and systematic error assessment. JMF and MR worked on the spectroscopic database. PR, MR, EC, BMD, LS, SC worked on the algorithm development. AD coordinated the whole MIPAS phase F project. AP, FB and MG worked on the quality control of all the data. BMD wrote the paper with feedback and contributions from all other co-authors. MG, PP and EP made all the figures of the paper.

*Competing interests.* The authors declare that they have no conflict of interest



*Acknowledgements.* The work described in this paper has been performed under the contract ESA ESRIN n. 4000112093/14/I/LG MIPAS and CCN2 'Support to MIPAS Level 2 Product Validation Phase F'. M.L.-P acknowledges financial support from the State Agency for Research of the Spanish MCIU through grant PID2019-110689RB-I00/AEI/10.13039/501100011033



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





**Table 1.** List of the MIPAS observation modes used for the level2-v8 database with their vertical coverage and number of available scans.

| MODE | Number of tangent altitudes | Altitude range of measurements | Number of available scans |
|---|---|---|---|
| FR Nominal | 17 | 6-68 | 510420 |
| OR Nominal | 27 | 6-70 | 1889082 |
| OR UTLS1 | 19 | 8.5-52 | 134638 |
| OR UTLS1-old | 18 | 8.5-49 | 10568 |
| OR UTLS2 | 11 | 12-42 | 5719 |
| OR AE | 12 | 7-38 | 4055 |
| OR NLC | 25 | 39-102 | 43541 |
| OR Upper Atmosphere | 35 | 42-172 | 166992 |
| OR Middle Atmosphere | 29 | 18-102 | 205784 |
| RR17 (Aug-Sep 2004) | 17 | 6-68 | 22768 |

**Table 2.** Size of the a priori errors and correlation lengths used in the retrievals .

| Target | Constant error [ ppmv] | Percentage error % | Correlation Length [km] |
|---|---|---|---|
| HCN | $10^{-6}$ | 90 | 6 |
| CFC$-14$ | 0.0 | 70 | 10 |
| COCl$_2$ | $10^{-7}$ | 95 | 6 |
| CH$_3$Cl | $10^{-6}$ | 95 | 4 (FR), 6 (OR) |
| C$_2$H$_2$ | $5 \times 10^{-8}$ | 90 | 4 |
| C$_2$H$_6$ | $10^{-7}$ | 90 | 5 |
| OCS | $10^{-7}$ | 80 | 5 |
| HDO | 0.001 | 100 | 10 |





**Table 3.** Average number of DOFs for the VMR profiles retrieved from scans not affected by clouds.

| Target | FR NOM | | OR NOM | |
|---|---|---|---|---|
| | tangent points | DOFs | tangent points | DOFs |
| Temperature | 17 | 14.5 | 27 | 20.2 |
| H2O | 17 | 11.9 | 27 | 14.2 |
| O3 | 17 | 16.7 | 27 | 23.82 |
| HNO3 | 13 | 12.6 | 18 | 15.8 |
| CH4 | 17 | 12.6 | 20 | 13.6 |
| N2O | 17 | 13. | 19 | 12. |
| NO2 | 15 | 10.7 | 16 | 9.5 |
| CFC−11 | 10 | 6.4 | 12 | 6.0 |
| CFC−12 | 12 | 8.3 | 16 | 8.3 |
| N2O5 | 10 | 6.0 | 15 | 6.6 |
| ClONO2 | 10 | 6.1 | 15 | 6.4 |
| CFC−14 | 14 | 11.7 | 11 | 9.5 |
| HCN | 16 | 8.3 | 13 | 6.4 |
| COF2 | 13 | 7.9 | 20 | 8.6 |
| CCl4 | 8 | 5.4 | 8 | 3.0 |
| HCFC−22 | 11 | 6.5 | 13 | 7.1 |
| C2H2 | 11 | 3.4 | 12 | 3.3 |
| CH3Cl | 11 | 3.0 | 15 | 2.4 |
| COCl2 | 11 | 4.2 | 21 | 3.6 |
| C2H6 | 13 | 2.9 | 15 | 1.8 |
| OCS | 17 | 4.7 | 19 | 4.2 |
| HDO | 17 | 9.7 | 24 | 9.0 |





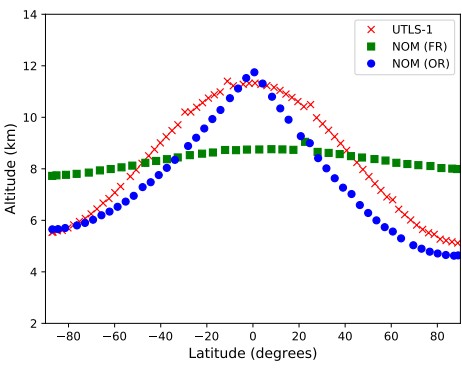

**Figure 1.** Altitude of the lowermost tangent point as a function of latitude for the NOM observation mode of the FR mission (green) and for the NOM (blue) and UTLS-1 (red) observation modes of the OR part of the mission.

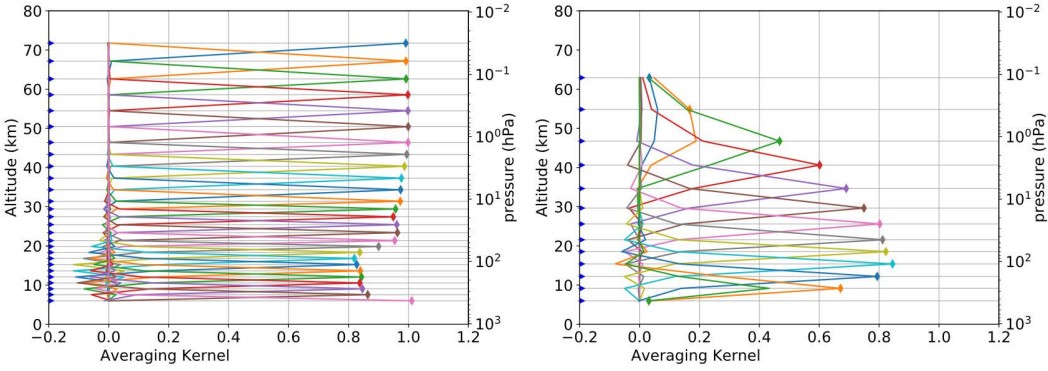

**Figure 2.** Examples of Averaging Kernels for the OR NOM observation mode: left panel for a species retrieved using a-posteriori regularisation (Ozone) and right panel for a species retrieved with OE (HCN)





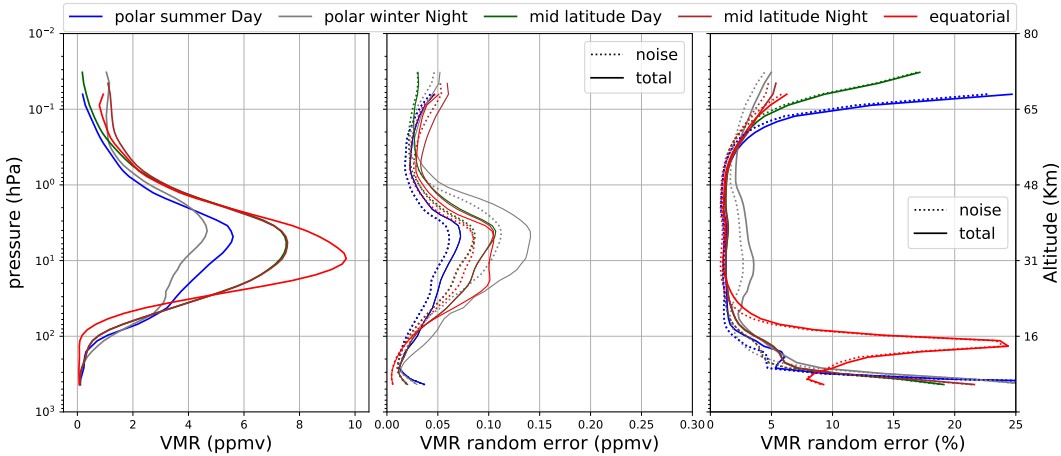

**Figure 3.** Ozone VMR for several conditions (left panel), absolute (middle panel) and relative (right panel) random error for the full resolution (FR) nominal mode averaged over 2003. The dotted lines represent the average noise error, the solid lines represent the total random error, coming from the quadratic summation of the noise error and the $p, T$ propagation error.

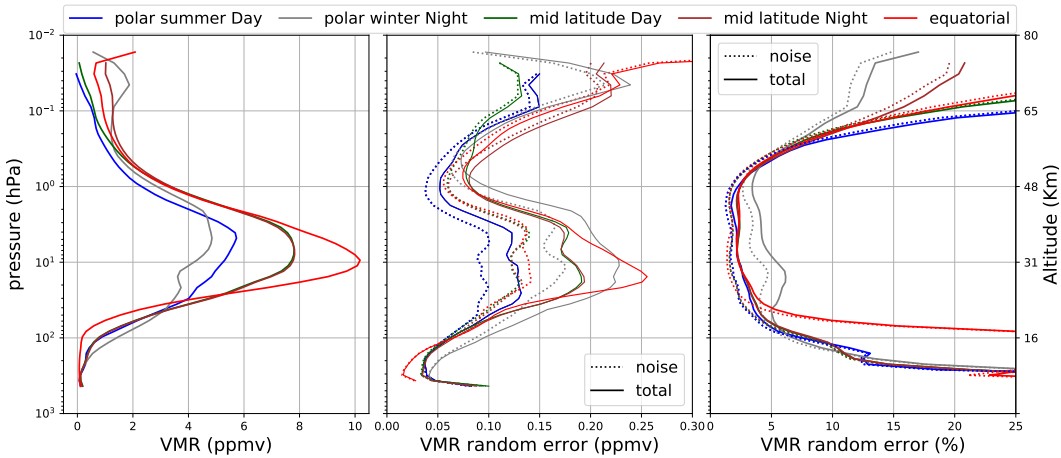

**Figure 4.** Ozone VMR for several conditions (left panel), absolute (middle panel) and relative (right panel) random error for the optimised resolution (OR) nominal mode averaged over 2010. The dotted lines represent the average noise error, the solid lines represent the total random error, coming from the quadratic summation of the noise error and the $p, T$ propagation error.





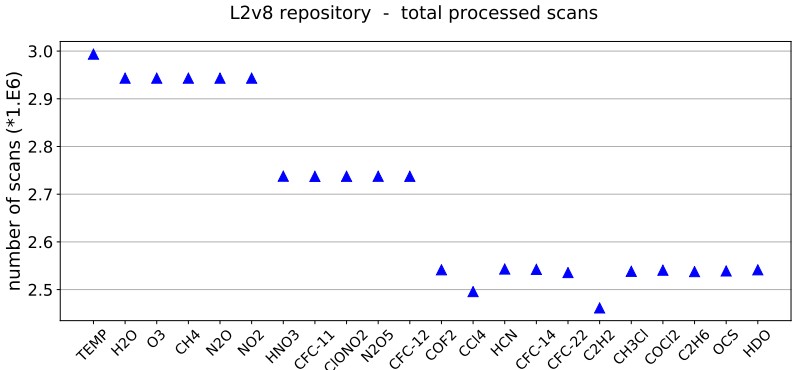

**Figure 5.** Total number of available profiles in the level2-v8 database for each species

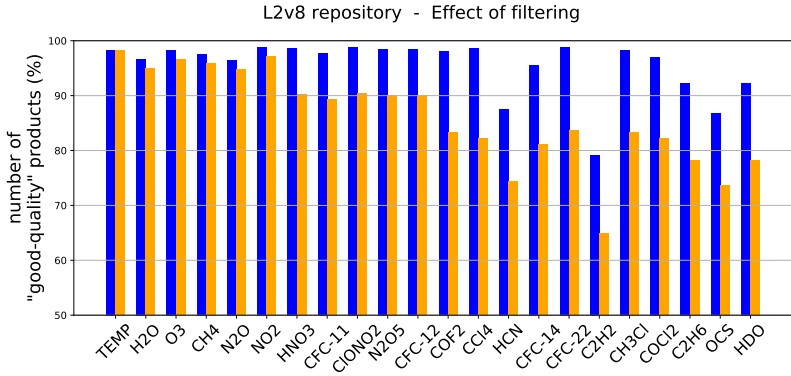

**Figure 6.** blue bars: percentage of 'good' profiles with respect to the profiles of the trace species contained in the MIPAS level2-v8 database; orange bars: percentage of 'good' profiles with respect to all the temperature profiles contained in the MIPAS level2-v8 database.





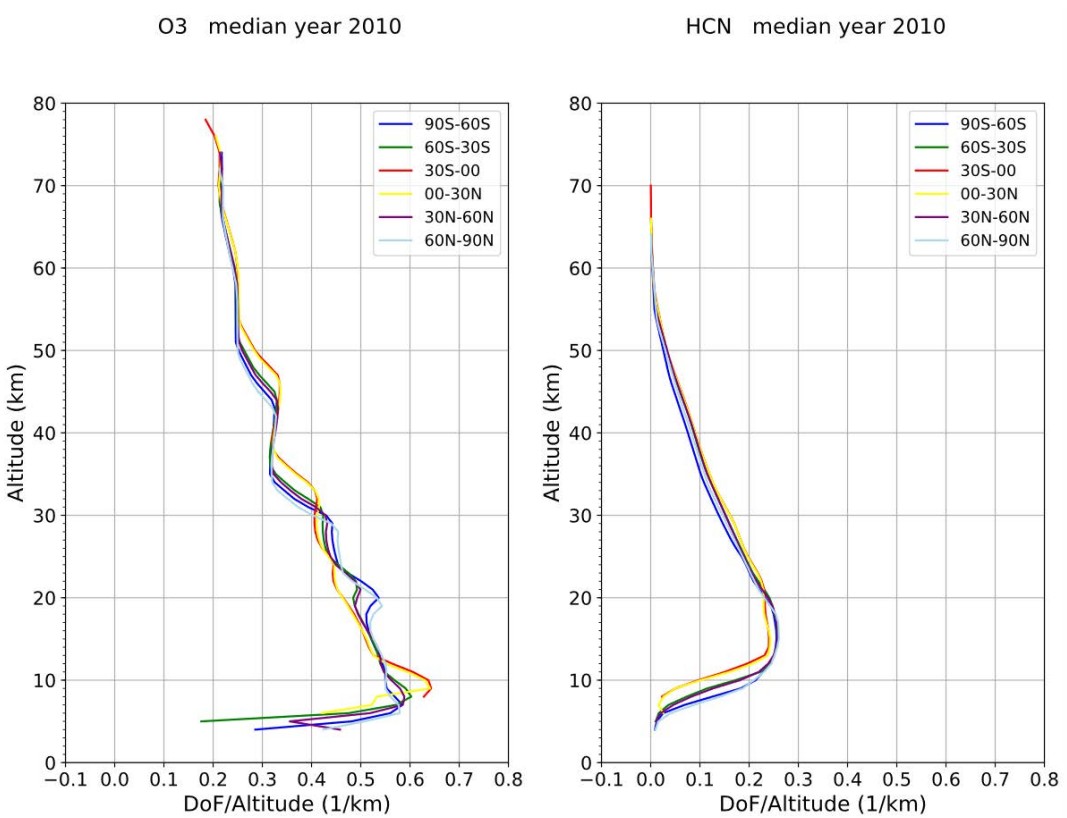

**Figure 7.** Examples of average profiles of DOF/altitude for a species retrieved using a-posteriori regularization (ozone) and for a species retrieved with OE (HCN).The different colours correspond to the different latitude bands as reported in the figures legends.

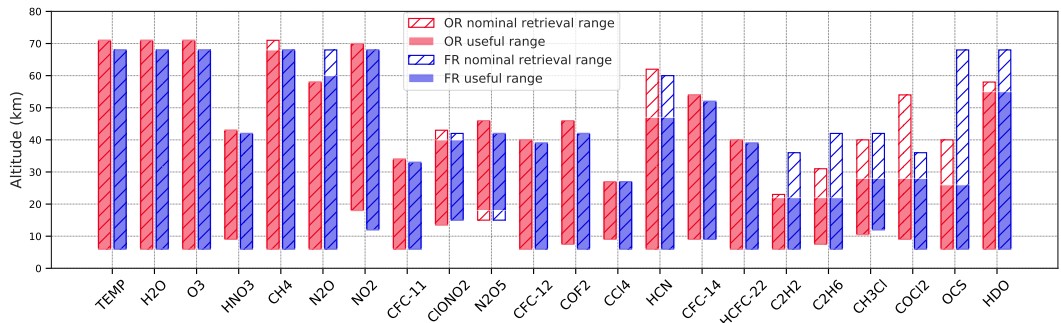

**Figure 8.** Retrieval range (open boxes) and useful range (shaded area) for all the molecules included in the level2-v8 database. Red indicates the OR period and blue the FR period.



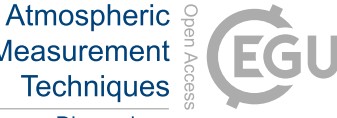



**Figure 9.** Number of temperature data present in the MIPAS level2-v8 database for each month and pressure bin, reported in six latitude bands. From top to bottom (left to right) Polar North (90N-60N), Mid Latitude North (60N-30N), Equatorial North (30N-0N), Equatorial South (0S-30S), Mid Latitude South (30S-60S), Polar South (60S-90S).







**Figure 10.** Time series of the monthly means of the temperature (left panels) and their random errors (right panels) as a function of pressure for the same latitude bands of figure 9. From top to bottom Polar North, Mid Latitude North, Equatorial North, Equatorial South, Mid Latitude South, Polar South.





**Figure 11.** Time series of the monthly means of the water vapor profiles (left panels) and their random errors (right panels) as a function of pressure for the same latitude bands of figure 9. From top to bottom Polar North, Mid Latitude North, Equatorial North, Equatorial South, Mid Latitude South, Polar South.





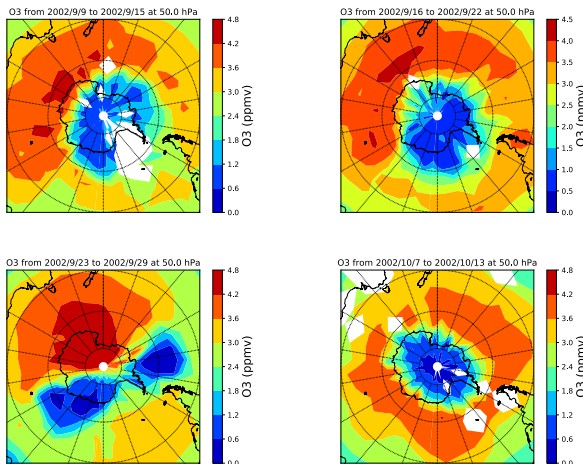

**Figure 12.** Distribution of the weekly means of ozone VMR at 50 hPa over the Southern polar region during the 2002 vortex split-up. Top-left panel 9-15 Sept, top right panel 16-22 Sept., left bottom panel 23-29 Sept and right bottom panel 7-13 Oct. 2002.

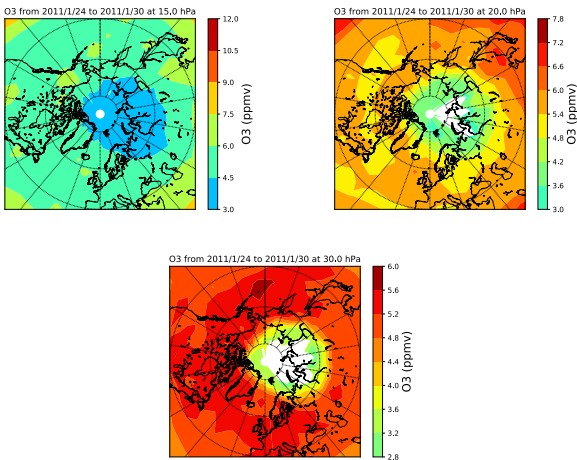

**Figure 13.** Distribution of the average ozone VMR at 15 (top left), 20 (top right) and 30 hPa (bottom) over the Northern polar region in the time period January $24^{th}$ to January $30^{th}$ during the winter 2011 ozone depletion. The white area is due to a lack of measurements caused by the presence of Polar Stratospheric Clouds.







**Figure 14.** Time series of ozone and HNO₃ in the Southern polar region along with the chlorinated species retrieved from MIPAS. While the time and pressure ranges are the same in each map, the colour scale (reported below each of them) is species dependent.

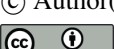

**Figure 15.** Time series of MIPAS level2-v8 database 'new' species in the northern mid-latitude region. Each species is plotted over its 'useful' vertical range. While the time and pressure ranges are the same in each map, the colour scale (reported below each of them) is species dependent.





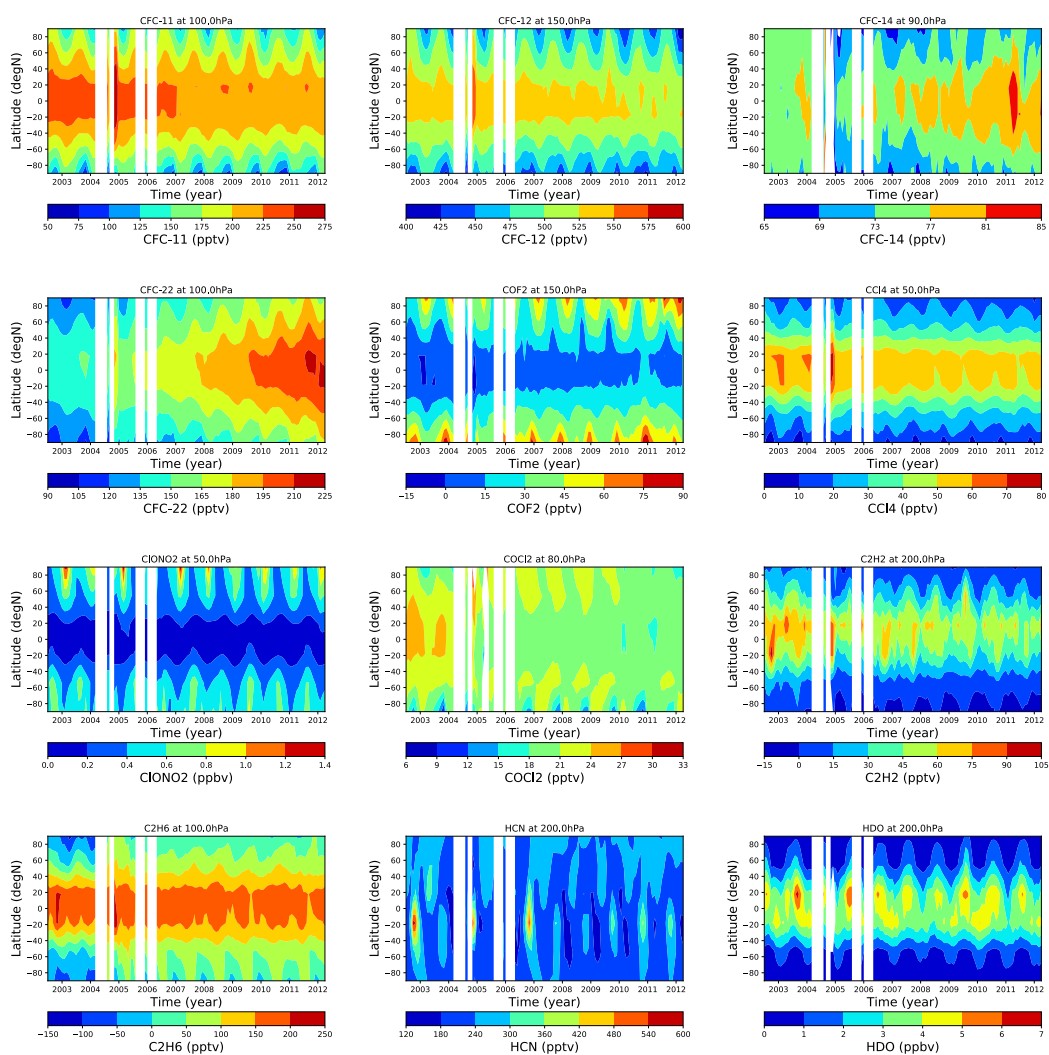

**Figure 16.** Time series of MIPAS level2-v8 database new species at the pressure levels where the maximum of the DOF/height curve is located.