# Peer review of "The ESA MIPAS/ENVISAT Level2-v8 dataset: 10 years of measurements retrieved with ORM v8.22"

_Atmospheric Measurement Techniques, 2021_

## Author Response (AR1)

**Answers to review 1**

**First of all we thank the reviewer for the suggestions. Below in italic the reviewer comments and in bold our answers**

*line 74—sp.  (Fisher → Fischer)*

**Already corrected in the on-line version of the manuscript**

*line 213—What is the nominal covered altitude range?*

**From 0 to about 65 km (pressure level 0.1 hPa). We will correct the text accordingly.**

Original text

Gradients are extended with 0 values outside the covered vertical range.

New text

Gradients are extended with 0 values outside the covered vertical range that ranges from 0km to about 65km (0.1 hPa).

*Section 6—The figures in this section are good and represent appropriate summaries of the dataset.*

**Thanks for this comment**

*Lines 505-507—It would be helpful to know (can you provide a reference or two?) whether or how scientific studies with MIPAS L2-v8 may differ or be improved from those already reported in the literature.*

**This is a difficult question, since we cannot predict how the scientific studies using the new MIPAS data could differ from the previous ones. However, the introduction of new species will surely trigger new investigations, like the one reported in the paper of Pettinari et al. (2021). The new database has been obtained using MIPAS level 1 data where in the calibration process the time dependent detector non linearities have been correctly taken into account, therefore we expect that trends estimated with this dataset will better represent the real atmospheric trends. We will add few sentences in the revised version of the paper**

Original text

The number of species included in the MIPAS level2-v8 dataset makes it of particular importance for the studies of stratospheric chemistry. The span of time covered by the database, and the accurate calibration of MIPAS measurements, enables studies of trends for the stratospheric and upper tropospheric concentration of the included gases. The data are reported at the vertical grid coincident with the tangent altitudes of the observations, spanning the altitude range from 6 to 68km in the FR and from 3 to 70km in the OR.

Modified text

The number of species included in the MIPAS level2-v8 dataset makes it of particular importance for the studies of

stratospheric chemistry. The span of time covered by the database, and the fact that the database has been developed using the last version of MIPAS level 1 data where the calibration errors due to time dependent detector nonlinearities have been reduced will ensure an improvement in the estimation of the trends for the stratospheric and upper tropospheric concentration of the included gases. An example of the studies that can be performed with the MIPAS level2-v8 database is reported in Pettinari et al. (2021), where the data of the new retrieved species $COCl_2$ have been used to study its trends in the UTLS. The data are reported at the vertical grid coincident with the tangent altitudes of the observations, spanning the altitude range from 6 to 68km in the FR and from 3 to 70km in the OR.

**Answers to review 2**

**First of all we thank the reviewer for the corrections and the suggestions on how to improve the paper. Below the reviewer comments in italic and our answers in bold.**

*GENERAL COMMENTS*
*================*

*The paper describes a new level 2 data set for the MIPAS instrument. All parts of the instrument, level 2 processing, and associated data products are well described in sufficient detail for the purpose of this paper.*
*The topic fits the journal, but the paper might be even better suited for ESSD (https://www.earth-system-science-data.net/).*

**In principle we agree with the reviewer, but at this stage of the publishing procedure we prefer to stay with AMT**

*I do not have any major criticism on this very nice paper, only a couple of minor issues mostly related to proper terminology and precise description.*

*I suggest that a language service should be used to review grammar and expressions; maybe the "normal" copy-editing by the journal is already sufficient for this purpose. I collected some suggestions for improvements as Minor Remarks.*

**We have implemented the suggestions, for the grammar we will wait for the copy-editing suggestions**

*The paper can be published after addressing the specific comments (a very minor revision).*

*None*

*line 194*

*--------*

*This and the preceding sentences sounds as if optimal estimation would be an alternative technique to Levenberg-Marquardt. According to mathematical literature, Levenberg-Marquardt is an algorithm for minimizing non-linear least squares problems (it's a special case of the trust region methods), whereas optimal estimation is a regularization scheme that can be seen as a special (or extended, depending on perspective) case of Tikhonov regularisation. That is, one can use the Levenberg-Marquardt algorithm to solve the non-linear minimisation required by an optimal estimation formulation of the problem.*

*Please clarify the text such that Levenberg-Marquardt is used in both cases for both problem formulations or clarify.*
*See also comment for line 269.*

**We agree, the reviewer is correct, we have reworded the text to avoid the confusion Regarding the "regularizing Levenberg-Marquardt" method used, please see also later, the answer to the comment about line 269.**

Original text

The majority of the retrievals are performed using the Levenberg-Marquardt technique (Levenberg, 1944; Marquardt, 1963), in order to cope with forward model non linearities, and an a-posteriori regularisation to avoid unphysical oscillations in the final results. For the molecules HCN, $CF_4$, $C_2H_2$, $C_2H_6$, $CH_3Cl$, OCS, $COCl_2$, and HDO, all characterised by very weak signals in MIPAS spectra, the OE technique has been used.

New text

All the retrievals are performed using the regularising Levenberg-Marquardt (LM) technique (Levenberg, 1944; Marquardt, 1963; Doicu et al., 2010) to cope with forward model non linearities. After convergence, an additional a-posteriori regularization is applied to avoid unphysical oscillations in the final profiles (Ceccherini, 2005; Ridolfi and Sgheri, 2009, 2011). For the molecules HCN, $CF_4$, $C_2H_2$, $C_2H_6$, $CH_3Cl$, OCS, $COCl_2$, and HDO, all characterised by very weak signals in MIPAS spectra, the OE technique has been used. The OE cost function is still minimised with the LM technique, and in this case the LM damping parameter is set to zero at the last iteration (see Raspollini et al., 2021).

*line 219*
* * *
*You use a constant a priori to capture the variability contained from the MIPAS information and not introduce any bias. Why did you deviate in the case of HDO from this principle?*

**This strategy has been adopted because the altitude of the hygropause can change significantly according to latitude and season, and a profile with a fixed shape, even if in pressure, was too far from reality to be a good a-priori estimate. A sentence to explain this has been added to the text.**

Original text

For HDO, the used a-priori information is the vertical distribution of $H_2O$ retrieved from the same scan, scaled by the natural $HDO/H_2O$ isotopic ratio.

New text

For HDO, the used a-priori information is the vertical distribution of $H_2O$ retrieved from the same scan, scaled by the natural $HDO/H_2O$ isotopic ratio. This strategy has been adopted because the altitude of the hygropause can change significantly according to latitude and season, and a profile with a fixed shape, even if in pressure, was too far from reality to be a good a-priori estimate.

*line 269*
* * *
*The Levenberg-Marquardt method is not a regularisation method, but an algorithm for identifying a (local) solution to (reasonably behaved) non-linear least-square problems. In contrast, regularisation approximates the original not well-behaved problem by a well-behaved one and thus typically leads to a different (but stable!) solution compared to the original problem (which is why one probably tries to have as "weak" as possible regularisation). When the Levenberg-Marquardt algorithm converges, it identifies a minimum to the original problem, not an approximate one and thus it is not a regularization by definition (see e.g. Engl, Hanke&Neubauer or Nocedal&Wright).*

*Please adopt the text to use mathematical standard terminology.*

**The method used is actually the "regularising Levenberg-Marquardt" (LM) described in Sect. 7.2.2 of the book of Doicu et al. 2010. In this method the iterations are stopped when a pre-defined convergence criterion is met. Iterations may stop also if the current value of the LM damping coefficient is not negligibly small. This implies that the method behaves like an iterative Tikhonov regularization using the zero-order derivative operator, and a-priori given by the state vector at the previous iteration. Generally, the regularisation obtained is very soft (small bias of retrieved profiles) and its actual strength is properly quantified by the averaging kernels, when they are evaluated with the correct iterative formula presented in Ceccherini and Ridolfi 2010. The online discussion of this paper (https://acp.copernicus.org/articles/10/3131/2010/acp-10-3131-2010-discussion.html )**
**contains further information on the characteristics of the approach.**

**In the revised paper we have introduced a reference to Doicu et al. 2010.**

Old text

For the species retrieved using the Levenberg-Marquardt regularizing method, the AKM is calculated taking into account all the steps performed during the retrieval iterations, as described in Ceccherini and Ridolfi (2010).

New text

For the species retrieved using the regularising Levenberg-Marquardt method, the AKM is calculated taking into account all the steps performed during the retrieval iterations, as described in Ceccherini and Ridolfi (2010).

*line 385*
* * *
*I am sure how to interpret this. When applying the AVK, e.g. to model data, one always needs to employ all columns of the AK matrix on the model data, whereas one may skip rows if the AKM associated with "not useful" elements of the profile. Is this what you are trying to express?*

**We wanted to say that the user should use all the terms of each averaging kernel and not only the ones relative to the vertical range that the user has selected. We have rephrased the sentence.**

*Further, you are using also non-zero a priori profiles (at least for OE). It would be useful to remind users here to not forget the (I-A) x_a term.*

**We have introduced the suggested recommendation in the revised text**

Original text

Since the AKM and the CM are calculated considering the retrieval 390 on the full vertical range, the use of only a part of the profile, with the association of sub-matrices of the provided AKM and CM, implies an approximation in the AKM and CM. Therefore it is recommended to use the profile, with its CM and AKM, on the full vertical range.

Revised text

Since the AKM and the CM are calculated considering the retrieval 390 on the full vertical range, even if some of the retrieved values are discarded by the user, we recommend to use the full profile along with its full CM and AKM. Moreover, for the species where OE is used, when using the AKM the user should take into account the adopted a-priori information (reported in the output files).

*lines 363, 366, 478, Figure 5, Figure 6*
* * *
*CFC-22 -> HCFC-22*

**Done, changed also in figure 16**

*MINOR REMARKS*
=============

***All the following suggestions have been accepted and implemented***

*line 67*
* * *
*time lapse -> period?*

*line 75*
* * *
*connections with -> connections to*

*line 83*
* * *
*a part from -> apart from*

*line 92*
* * *
*to the -> to*

*line 102*
* * *
*the "." should be on the end of the preceding line*
*line 176*
*--------*
*Section -> section*

*line 184*
*--------*
*HDO -> and HDO*

*line 232*
*--------*
*Table numbers are missing (??)*

*line 262*
*--------*
*netcdf -> NetCDF; also in the following*

*line 273*
*--------*
*AKs; abbreviation not introduced*

*line 542*
*--------*
*"are thought for" -> "are for" ?*